# Narrative review after post-hoc trial analysis of factors that predict corneal endothelial cell loss after phacoemulsification: Tips for improving cataract surgery research

**Jean-Marc Perone**[1]*, **Marie-Soline Luc**[1], **Yinka Zevering**[2], **Jean-Charles Vermion**[1], **Grace Gan**[1], **Christophe Goetz**[2]

**1** Ophthalmology Department, Regional Hospital Center of Metz-Thionville, Mercy Hospital, Metz, Grand Est, France, **2** Clinical Research Support Unit, Regional Hospital Center of Metz-Thionville, Mercy Hospital, Metz, Grand Est, France

* jm.perone@chr-metz-thionville.fr

## Abstract

### Purpose

Identifying pre/perioperative factors that predict corneal endothelial-cell loss (ECL) after phacoemulsification may reveal ways to reduce ECL. Our literature analysis showed that 37 studies have investigated one or several such factors but all have significant limitations. Therefore, the data of a large randomized controlled trial (PERCEPOLIS) were subjected to post-hoc multivariate analysis determining the ability of nine pre/perioperative variables to predict ECL.

### Methods

PERCEPOLIS was conducted in 2015–2016 to compare two phacoemulsification techniques (subluxation and divide-and-conquer) in terms of 3-month ECL. Non-inferiority between the techniques was found. In the present study, post-hoc univariate and multivariate analyses were conducted to determine associations between ECL and age, sex, cataract density, preoperative endothelial-cell density, phacoemulsification technique, effective phaco time (EPT), and 2-hour central-corneal thickness. The data are presented in the context of a narrative review of the literature.

### Results

Three-month data were available for 275 patients (94% of the randomized cohort; mean age, 74 years; 58% women). Mean LOCSIII cataract grade was 3.2. Mean EPT was 6 seconds. Mean ECL was 13%. Only an older age (beta = 0.2%, p = 0.049) and higher EPT (beta = 1.2%, p = 0.0002) predicted 3-month ECL. Cataract density was significant on univariate (p = 0.04) but not multivariate analysis. The other variables did not associate with ECL.

**Data Availability Statement:** The datasets generated during and/or analyzed during the current study are not publicly available according

to French Law No. 2018-493 of June 20, 2018 on the protection of personal data (The General Data Protection Regulation (Regulation (EU) 2016/679) (GDPR: article 9) but are available from the Clinical Research Support Platform (Plateforme d'Appui à la Recherche Clinique [PARC]) of the Regional Central Hospital (CHR) of Metz-Thionville on reasonable request (email: projetrecherche@chrmetz-thionville.fr; tel: +33 3 87 17 98 82). All non-archived data is subject to daily backups while all archived data is subject to duplicate storage at two different sites. This data processing is compliant with a baseline reference methodology (MR-004) to which the CHR Metz-Thionville signed a compliance commitment on October 8, 2018.

**Funding:** The author(s) received no specific funding for this work.

**Competing interests:** The authors have declared that no competing interests exist.

## Conclusions

Older age may amplify ECL due to increased endothelial cell fragility. EPT may promote ECL *via* cataract density-dependent and -independent mechanisms that should be considered in future phacoemulsification research aiming to reduce ECL. Our literature analysis showed that the average ECL for relatively unselected consecutively-sampled cohorts is 12%.

## Introduction

A relatively common iatrogenic complication of cataract surgery is pseudophakic bullous keratopathy (PBK), which emerges 8 months to 7 years after surgery [1]. It is caused by destruction of the corneal endothelial cells by the heat, free radicals, and fluid turbulence generated during phacoemulsification [2–4]: when endothelial-cell numbers drop below 300–500 cells/mm$^2$ [5], the corneal pump function is impaired, causing fluid to accumulate. This results in irreversible diffuse edema that increases corneal thickness, decreases corneal transparency, and results in an irregular corneal surface, thereby significantly impairing visual acuity [6]. While the risk of PBK has decreased in the last few decades due to advances in cataract surgery and ophthalmic viscosurgical devices (OVDs) [7, 8], it still occurs in 0.1–2% cataract surgeries [3, 4]. Since 10 million cataract operations are conducted annually [9], PBK poses a significant medical burden.

Identifying preoperative or surgical risk factors for endothelial-cell loss (ECL) after cataract surgery could improve outcomes: it could help determine which patients will require additional care or alternative treatment strategies and/or promote the development of new endothelial cell-sparing technical approaches. Our comprehensive analysis of the literature published since 1995 revealed a large field composed of 37 studies that have searched for such risk factors (Table 1) [8, 10–45]. However, all have significant limitations. First, all but six (84%) looked at fewer than 100 eyes and/or examined ECL before 3 postoperative months, which is when ECL may still be stabilizing [12, 16, 18, 33]. Second, of the remaining six studies, four only conducted univariate analyses [8, 14, 27, 31] and the two multivariate analyses were performed in 1996 and 2004, respectively [41, 43]. Since phacoemulsification technology has progressed considerably in the last two decades [46], analyses of more recent large cohorts are warranted.

We recently conducted the PERCEPOLIS randomized controlled trial (RCT) (n = 292) to compare ECL after two phacoemulsification techniques, namely, subluxation and divide-and-conquer (DAC). Subluxation is a supracapsular technique where the nucleus is transposed into the anterior chamber and then fragmented [47, 48]. DAC is an endocapsular method where the nucleus is fractured into four quadrants in the capsular bag and then removed [49]. Our RCT showed that subluxation is non-inferior to DAC in terms of ECL at 3 months [50]. In the present study, we conducted a post-hoc analysis of these trial data to assess the ability of nine demographic, preoperative, and perioperative variables to predict ECL 3 months after phacoemulsification. Both univariate and multivariate analyses were conducted and the findings are presented in the context of a narrative review of the literature.

## Patients and methods

### Study design and ethics

PERCEPOLIS (PERte Cellulaire Endotheliale après PhacOemuLsification Intra ou Supracapsulaire/endothelial cell loss after endo- or supracapsular phacoemulsification) is a single-center

**Table 1. Studies that examined associations between phacoemulsification-induced ECL and pre/perioperative factors in cohorts similar to ours.**

| Ref no. date | Study type | Special cohort characteristic[a] | Mean (range) nuclear sclerosis grade, [excluded grades][b] | No. eyes | Postop ECLtimept | ECL (%)[c] | Type analysis | Older age | Sex | Higher cataract density | Lower preop ECD | Shallower preop ACD | Longer phaco time | More phaco energy[d] | Higher EPT (or EFX[e]) | Higher CDE | Longer surgery time | Postop CCT |
|---|---|---|---|---|---|---|---|---|---|---|---|---|---|---|---|---|---|---|
| Present study | RCT | | Mod-hard 3.2 (1–5) [b/w cat] | 275 | 3m | 13 | UV | 0.001 | NS | 0.04 | NS | NS | NS | 0.001 | 0.001 | | NS | NS |
| | | | | | | | MV | 0.035 | NS | NS | NS | NS | | | 0.001 | | | NS |
| Ma [10] 2021 | Pros | | Mild 1.4 (1–2) [≥3] | 32 | 6m | 6 | UV | | | | | | NS | | | 0.01 | | |
| Bu [11] 2021 | Pros | | 2–4 | 148 | 1m | | UV | | | | | 0.01 | | | | | | |
| Bea [12] 2021 | C-C | Age/sex-mtc[f] | Mild 1.9 (1–3) [b/w] | 43 | 6m | 21 | MV | 0.042 | NS | 0.001 | NS | NS | | | | | | |
| Joo [13] 2021 | Pros | | Mod 2.7 (2–3) | 37 | 12m | 11 | UV | NS | | | | | | | | | | |
| Dz [14] 2020 | RCT | | Mod 2.6 (1–4) | 134 | 3m | 8 | UV | | | | | | | | NS | 0.01 | | |
| Choi [15] 2019 | Retr | Phaco>10 y | Mild-mod 1.6 (0–3) Emery | 81 | >10y | | UV | NS | NS | <0.001 | NS | NS | | | | | | 0.001 |
| | | | | | | | MV | | | <0.001 | | | | | | | | 0.001 |
| Kra [16] 2019 | RCT | | Mod 2.1 (2–3) | 96 | 6m | 17 | UV | | | NS | 0.023 | | | | | 0.0001 | NS | |
| | | | | | | | MV | | | NS | 0.006 | | | | | 0.0001 | | |
| Ga [17] 2019 | C-C | | Unclear | 80 | 3m | 18 | MV | 0.03 | NS | | | | | | 0.002 | | | |
| Per [18] 2018 | Pros | | Up to 4 [white] | 85 | 1m | | UV | | | | | | | | | | | 0.001 |
| Al [19] 2018 | Pros | | 13.9 (7.6–22.6) Scheim | 62 | 1m | | UV | | | 0.001 | | | | | | | | |
| Sin [20] 2017 | RCT | Hard cat | Hard (3–4) [≤2] | 152 | 1.5m | | UV | 0.01 | | 0.01 | | 0.001 | | | 0.007 | | | |
| Mis [21] 2015 | Pros | | Unclear | 23 | 3m | 9 | UV | | | | | | | | 0.002 | | | |
| Me [22] 2015 | Pros | | Mod 2.5 (1–4) | 500 | 1.5m | | UV | | | r = 0.98 | | | | | | | | |
| Ata [23] 2014 | Pros | | Mod 2.7 (1–4) | 86 | 1m | | UV | | | 0.001 | | | | 0.007 | 0.022 | 0.028 | | |
| Ma [24] 2014 | Retr | | Mild-mod 2.0 PNS | 62 | 1m | | UV | | | 0.0001 | | | | | | | | |
| Ors [25] 2014 | Retr | | Mod 41% grd 3 (1–6) | 365 | 1m | | UV | 0.01 | | 0.001 | | | | | | | | |
| Co [26] 2013 | RCT | | Mod-hard 3.1 (1–4+) | 73 | 3m | 14 | UV | | | | | | | | r = 0.43 | | | |

(Continued)

Table 1. (Continued)

| Ref no. date | Study type | Special cohort characteristic[a] | Mean (range) nuclear sclerosis grade, [excluded grades][b] | No. eyes | Postop ECLtimept | ECL (%)[c] | Type analysis | Older age | Sex | Higher cataract density | Lower preop ECD | Shallower preop ACD | Longer phaco time | More phaco energy[d] | Higher EPT (or EFX[e]) | Higher CDE | Longer surgery time | Postop CCT |
|---|---|---|---|---|---|---|---|---|---|---|---|---|---|---|---|---|---|---|
| Sol [27] 2012 | Pros | | Mod-hard 3.1 (1–5) | 120 | 3m | 15 | UV | NS | | K = 0.42 | | | | | | r = 0.43 | | |
| Go [28] 2012 | RCT | Hard cat | Hard 4.4 (4–5) | 70 | 3m | | UV | | | | | | 0.001 | | | 0.001 | | |
| Tak [29] 2012 | Pros | | Mod 2.1 (1–3) [4+] PNS | 38 | 1m | | UV | | | | | | | | | | | r = 0.56 |
| Fara [30] 2011 | RCT | | Mod | 30 | 3m | 14 | UV | [0.07] | | | | NS | | | 0.001 | | | |
| Ma [31] 2011 | C-C | Age-mtc[f] | Mild-mod 2.0 (1–5) | 158 | 3m | 17 | UV | | | NS | | | | | | | NS | |
| Luc [32] 2011 | RCT | | Mild-mod 2.0,2.3(1–4) | 50 | 2m | | UV | | | | | | 0.0001 | | | | | |
| Reu [8] 2010 | RCT | | Mod-hard 3–4 | 182 | 3m | 7 | UV | | | | | NS | NS | | | NS | | |
| Cho [33] 2010 | Pros | | 1–4 | 94 | 3m | 17 | MV | | | 0.001[g] | | 0.000 | | | NS[g] | | | |
| Cho2 [34] 2010 | Pros | | Mod-hard 3.1 (1–5) | 88 | 2m | | UV | | | 0.000 | | 0.04 | | | | | | |
| | | | | | | | MV | | | 0.000 | | NS | | | | | | |
| Bar [35] 2009 | RCT | | Mod-hard 3+ | 60 | 3m | 9 | MV | NS | NS | | <0.001 | | | | <0.001 | | | |
| Lee [36] 2009 | RCT | | Mod 2–4 | 43 | 2m | | MV | 0.006 | | 0.043 | NS | | | | | NS | | |
| Sto [37] 2008 | RCT | | Mod 2.8, 2.9 Emery | 60 | 12m | 5 | UV | NS | | NS | NS | NS | | NS | | | | |
| Per [38] 2006 | RCT | | Mod-hard 3–4 | 50 | 3m | 9 | UV | NS | | | NS | NS | NS | | | | NS | |
| Lundber [39] 2005 | Pros | ≤5%, 6–20%, ≥21% d1 CCT | Mod-hard 3.1 (1–5) | 30 | 3m | 17 | UV | [0.098] | | 0.001 | NS | NS | 0.023 | | NS | NS | 0.07 | 0.0001 |
| | | | | | | | MV | NS | | 0.024 | | | NS | | NS | NS | NS | NS |
| O'Brien [40] 2004 | Pros | 30% w/b cat, junior doctor | Mod 2.8 (1–4) Emery | 40 | 1m | | UV | NS | | 0.024 | NS | NS | 0.0004 | 0.008 | 0.003 | | 0.03 | |
| | | | | | | | MV | | | 0.048 | | | 0.0004 | NS | NS | | NS | |
| Bo [41] 2004 | RCT | Half ECCE | Mod 91% 1–3 [grd 5] | 433 | 12m | 11[h] | MV | 0.004 | NS | 0.025 | NS | | | | | | | |
| Walk [42] 2000 | Pros | | Not stated | 50 | 12m | 9 | UV | NS | | | | 0.001 | 0.01 | NS | | | | |
| | | | | | | | MV | NS | | | | NS | 0.003 | | | | NS | |

(Continued)

**Table 1.** (Continued)

| Ref no. date | Study type | Special cohort characteristic[a] | Mean (range) nuclear sclerosis grade, [excluded grades][b] | No. eyes | Postop ECLtimept | ECL (%)[c] | Type analysis | Older age | Sex | Higher cataract density | Lower preop ECD | Shallower preop ACD | Longer phaco time | More phaco energy[d] | Higher EPT (or EFX[e]) | Higher CDE | Longer surgery time | Postop CCT |
|---|---|---|---|---|---|---|---|---|---|---|---|---|---|---|---|---|---|---|
| Hay [43] 1996 | Pros | | Mod 1–5 (2 most often) Emery | 859 | 3m | | UV | 0.0001 | NS | 0.0001 | | | | 0.001 | | | | |
| | | | | | | | MV | | NS | 0.0001 | | | | NS | | | | |
| Dic [44] 1996 | RCT | | Not stated | 58 | 12m | 7 | UV | | | | | | Linear | Linear | | | | |
| Zet [45] 1995 | CS | | Not stated | 64 | 3m | 4 | UV | | | | | | | NS | | | | |

[a] All cohorts employed in the univariate or multivariate analysis were selected using standard eligibility criteria (i.e. senile cataract, no other ocular diseases) but some also had the indicated characteristics.

[b] All LOCSIII except where indicated otherwise.

[c] The ECLs of studies on cohorts that did not resemble ours are not shown. These studies included eyes with low preoperative ECD, postoperative ECL measurement timepoints less or more than 3–12 months, focused on eyes with very hard cataracts, did not report ECL data, or did not report when ECL was measured.

[d] Phaco energy was expressed as ultrasound power, total phaco energy, average ultrasound power %, phaco energy, and total ultrasound energy.

[e] EFX = EPT measured on a WhiteStar Signature machine multiplied by a specific coefficient for the transversal movement; expressed in seconds.

[f] Control group for a study on a cohort with diabetes mellitus.

[g] This study comprised three groups: eyes with shallow, moderate, and deep ACDs. Multivariate analyses were conducted in each group with both cataract density and EPT. ECL in the shallow and deep groups were predicted by cataract density but not EPT. By contrast, ECL in the moderate group was predicted by EPT but not cataract density.

[h] ECL of phacoemulsification group only.

ACD, anterior chamber depth; b/w cat, brown/white cataracts; cat, cataract; C-C, case-control study; CCT, central corneal thickness; CDE, cumulative dissipated energy; CS, case series; d1 CCT, CCT on postoperative day 1; ECCE, extracapsular cataract extraction; ECD, endothelial cell density; ECL, endothelial cell loss; Emery, Emery–Little classification; EPT or EFX, effective phaco time; K, Kendall's tau; Linear, linear relationship but correlation coefficient not reported; LOCS, Lens Opacity Classification System; m, months; mtc, matched; mod, moderate; MV, multivariate analysis; NS, not significant; phaco, phacoemulsification; PNS, Pentacam Nucleus Staging system; postop, postoperative; Pros, prospective study; r, correlation coefficient or Spearman's rho; RCT, randomized controlled trial; Retro, retrospective study; Scheim, Scheimpflug cataract density measurement; stratif, cohort selected with stratification for the indicated variables; timept, timepoint; UV, univariate analysis; y, year

parallel-arm interventional RCT (ClinicalTrials.gov Identifier: NCT02535819, IDRCB 2015-A00789-49) that was conducted in 4 June 2015–4 April 2016 at the Metz-Thionville Regional Hospital Center (Metz, France). It was approved by the Ethics Committee of the French Society of Ophthalmology (IRB 00008855 Société Française d'Ophtalmologie IRB#1) and adhered to the tenets of the Declaration of Helsinki. All subjects provided written informed consent before randomization [50].

## Patient cohort

The details of PERCEPOLIS have been published [50]. Briefly, the cohort consisted of a convenience series of 292 adult patients who had a nuclear (NO1–NO4; NC1–NC4), cortical (C1–C5), or posterior subcapsular (P1–P5) cataract, as determined by using the Lens Opacities Classification System (LOCS)III classification [51], and a best spectacle-corrected visual acuity (BSCVA) of >+0.2 logMAR and were randomized to undergo either subluxation ($n$ = 148) or DAC ($n$ = 144). Exclusion criteria were: white/brown cataracts; insulin-dependent diabetes or diabetic retinopathy; preexisting cornea pathology; intraocular-pressure (IOP) pathology; posterior-segment pathology; preoperative endothelial-cell density (ECD) <1500 cells/mm$^2$; pregnancy; and history of retinal detachment, ocular trauma, or anterior/posterior-segment surgery. Patients undergoing additional procedures apart from cataract removal and intraocular-lens implantation were also excluded. Only first operated eyes were included in the study.

Although ECD and other data were collected from the PERCEPOLIS cohort at 1, 3, and 12 months, we chose the 3-month data for the present study because there was a large loss to follow-up at 12 months (32%), meaning there may have been some selection of the 12-month patients. Moreover, ECL is generally considered to have stabilized at 3 months [12, 16, 18, 33, 52]. Three-month ECD data were available for 275 of the 292 cohort eyes (94%). These 275 eyes formed the cohort for the present post-hoc analysis.

## Phacoemulsification surgery

All surgeries were performed by one experienced surgeon (J-MP). As described previously [50], the patients were given topical anesthesia, pupillary dilation was induced, a blepharostat was positioned, a coaxial 2.2-mm corneal mini-incision was created, dispersive viscoelastic (DuoVisc; Alcon Laboratories, Switzerland) was injected into the anterior chamber, a second incision (90˚ to the first incision) was generated with a 20-gauge needle, circular capsulorhexis was conducted, and lens-nucleus hydrodissection was performed with either DAC [49] or subluxation [47, 48]. All phacoemulsifications were performed with the same machine (Stellaris; Bausch & Lomb, Inc., Canada) with the following parameters: 450 mmHg vacuum, 35 cc/min fixed aspiration rate, 110-cm bottle height, and 60–40% phacopower. After phacoemulsification, an acrylic intraocular lens was implanted, viscoelastic was completely removed with irrigation and aspiration, and the corneal incision was gently hydrated. Surgery was completed with a corneal stitch if required and patients were given an intracameral injection of cefuroxime (Aprokam; Laboratoires Théa, France).

## Variables collected

The following data were gathered from the PERCEPOLIS trial database because they were available and could potentially predict final ECL: patient age, sex, and cataract density; preoperative anterior-chamber depth (ACD), central-corneal thickness (CCT), and central ECD; and surgery type (DAC vs. subluxation), surgery time, average phaco power %, absolute phaco time (APT), and effective phaco time (EPT). CCT 2 hours after surgery was also collected because several studies have suggested early CCT can predict final ECL after

phacoemulsification [15, 18, 29, 39]. In 15% of the present cohort, 2-hour CCT could not be determined because the patients did not attend the consultation. Treated eye side and preoperative IOP were also collected as cohort descriptors. ECD measured 3 months after surgery was used to calculate ECL at 3 months relative to baseline. The 3-month timepoint was chosen for analysis rather than the 12-month timepoint in our trial because loss to follow-up was 6% at 3 months and 29% at 12 months. Cataract density was expressed as nuclear color (NC) and nuclear opacity (NO) and generalized as nuclear sclerosis (NS) since NC and NO correlate closely with each other and with cataract hardness [53–55]. ACD was measured with IOLMaster 4 (Carl Zeiss Meditec, Inc., Dublin, CA). Central ECD and CCT were measured with non-contact specular microscopy (CEM-530, Nidek Co., Ltd., Gamagori, Japan). APT corresponds to the time in seconds when the phaco power is on. EPT is the time in seconds when the ultrasound is at 100% power (calculated as APT × average phaco power %) [56]. Surgery time was monitored from incision to final control of corneal sealing with a chronometer. Ultrasound power %, APT, and EPT were recorded from the phacoemulsification device at the end of surgery.

## Statistical analysis

The per-protocol trial data were analyzed: 15 of the 292 phacoemulsifications had been converted to the other method for technical reasons (an overly soft core, poor pharmacologically induced pupillary dilation, significant anterior chamber narrowness, and severely hard crystalline lens). However, intention-to-treat analysis of the trial data also showed that subluxation was non-inferior to DAC [50]. All variables are presented as mean±standard deviation (SD) or n (%). Correlations between 3-month ECL and continuous pre/perioperative variables were determined with Pearson correlation analyses. Categorical groups were compared in terms of 3-month ECL by Student's $t$-test or ANOVA. Eyes that did/did not have 2-hour CCT data were compared in terms of pre/perioperative variables and ECL by Student's $t$-test or Fisher's exact test. Multiple linear regression analysis was conducted to identify which of nine variables can independently predict ECL at 3 months. Given the cohort sample size (n = 275) and the 1 in 10 rule of thumb in multivariate analysis that states one candidate predictor can be studied for every 10 patients [57], the sample size was ample for this analysis. Before this, collinearity between the variables was assessed by determining Variance Inflation Factors and condition indices. Ultrasound power and APT were collinear with EPT because EPT is calculated on the basis of both (EPT = APT × mean ultrasound power %) [56]. Thus, ultrasound power and APT were excluded from multivariate analysis. Treated eye side and IOP were also not included in multivariate analysis. Cataract density either served as a categorical variable or was forced as a quantitative variable, with similar results. Missing 2-hour CCT values were ignored during multivariate but not univariate analysis. All analyses were performed in SAS (version 9.3, SAS Inst., Cary, NC, USA). P values <0.05 indicated statistical significance.

## Identification of similar studies in the literature for narrative review

Given the large body of related previous research, it was necessary to conduct a comprehensive literature search for similar studies. The study-inclusion criteria were: (i) study was published after 1995; (ii) study employed univariate and/or multivariate analyses to identify associations between ECL and one or more of our nine pre/perioperative variables; (iii) the cohort generally resembled ours: the patients had senile cataract, were consecutive or randomly selected, lacked other ocular pathologies and diabetes, and there were no or few white/brown cataracts; and (iv) phacoemulsification was conducted by one or more senior surgeons with standard phacoemulsification procedures. The latter were defined as longitudinal, torsional, or

transversal phacoemulsification [femtosecond laser-assisted cataract surgery (FLACS) was excluded] with any machine, incision size ≥2.2 mm, any OVD, and any phacoemulsification technique. Two early studies (2004 and 2005) whose cohorts did not fully meet our inclusion criteria were included because they were multivariate analyses on a wide range of the preoperative/surgical variables and are often cited in the literature: in O'Brien et al. [40], the surgeon was a junior resident and 30% of the study eyes had white/brown cataracts while in Lundberg et al. [39], the patients were selected for differing degrees of day-1 CCT change. Our inclusion criteria also meant that in studies with a diabetic group or non-standard phacoemulsification procedure, we only included the data of the non-diabetic/conventional-phacoemulsification group. Several studies that asked a specific question and therefore imposed more specific patient selection criteria (e.g. low preoperative ECD [58–60], different ACDs [58, 61–64], occludable angles [65], or pseudoexfoliation syndrome [66, 67]) were excluded. S1 and S2 Tables show the details of the included studies.

## Results

Of the 292 study patients who were treated with phacoemulsification, 275 attended the 3-month PERCEPOLIS visit (94%). Of the 17 patients who did not attend the 3-month study visit, seven were lost to follow-up, eight withdrew consent, one underwent an implant change, and one missed the 3-month visit [50].

The 3-month visit attendees were on average 74 years old and 58% were women. The right eye was treated in 51% of patients. Preoperative IOP was 17 mmHg. Cataract density was normal (NS1), mild (NS2), moderate (NS3), moderately-severe (NS4), and severe (NS5) in 1%, 16%, 49%, 33%, and 1%, respectively. Thus, 83% had moderate-severe cataracts. Mean LOC-SIII score was 3.2. Mean preoperative ACD was 3.1 mm. Subluxation and DAC were used in 127 (54%) and 107 eyes (46%), respectively. Average surgical time, ultrasound power, APT, and EPT were 5:48 minutes, 15%, 40 seconds, and 6 seconds, respectively (Table 2).

The 2-hour CCT data were available for 234 eyes (85%). The eyes that did and did not have 2-hour CCT data did not differ significantly in terms of pre/perioperative variables (including preoperative CCT) or ECL (S3 Table). Average preoperative CCT of the $n = 234$ and total $n = 275$ eye cohorts was 560 and 561 μm, respectively. CCT rose to 584 μm at 2 hours (Table 2 and S3 Table).

Mean preoperative ECD was 2386 cells/mm$^2$, and this dropped to 2070 cells/mm$^2$ at 3 postoperative months. Thus, ECL at 3 months was 13% (Table 2).

Univariate analyses showed that higher ECL at 3 months associated with older age (r = 0.27, p<0.001), greater cataract density (8% for NS1/2 *vs.* 13–15% for higher grades, p = 0.04), and more ultrasound power (r = 0.25, p<0.001), APT (r = 0.24, p<0.001), and EPT (r = 0.36, p<0.001). The other variables, including 2-hour CCT, did not associate with ECL (Table 3).

To determine whether these variables could predict 3-month ECL, we conducted multiple linear regression analysis. Ultrasound power and APT were not included in this analysis because EPT is calculated on the basis of both [56] and thus correlates closely with both. Since there was no collinearity between age, cataract hardness, EPT, or the other six variables, they were all included in the regression analysis. Only an older age (beta = 0.2%, p = 0.049) and higher EPT (beta = 1.2%, p = 0.0002) predicted 3-month ECL (Table 4). Thus, ECL increased by 0.2% for every additional year of patient age and 1.2% for every second longer EPT. The model explained 15.7% of the variance in ECL (p<0.0001).

## Discussion

This study showed that of nine pre/perioperative variables, only older age and higher EPT were independent predictors of greater ECL after cataract surgery.

**Table 2. Demographic and operative characteristics of the study cohort.**

| Variable† | Average±SD | n (%) |
|---|---|---|
| Age, years | 74±9 | |
| Female sex | | 159 (58) |
| Right eye treated | | 140 (51) |
| Intraocular pressure, mmHg | 17±3 | |
| Cataract density, LOCSIII grade | | |
| NS1 | | 3 (1) |
| NS2 | | 45 (16) |
| NS3 | | 134 (49) |
| NS4 | | 91 (33) |
| NS5 | | 2 (1) |
| Mean LOCSIII NS score | 3.2±0.7 | |
| ACD, mm | 3.1±0.4 | |
| CCT, μm | | |
| Preoperative | 560±36 | |
| 2-hour postoperative (n = 234) | 584±134 | |
| ECD, cells/mm$^2$ | | |
| Preoperative | 2386±286 | |
| Postoperative month 3 | 2070±430 | |
| ECL | 13.0% | |
| Surgical technique | | |
| Subluxation | | 144 (52) |
| Divide-and-conquer | | 131 (48) |
| Surgery time, min:sec | 5:48±1.50 | |
| Ultrasound power (%) | 15±6 | |
| APT, seconds | 40±14 | |
| EPT, seconds | 6±3 | |

† n = 275 unless otherwise indicated

ACD, anterior chamber depth; APT, absolute phaco time; CCT, central corneal thickness; ECD, endothelial cell density; ECL, endothelial cell loss; EPT, effective phaco time; LOCSIII, Lens Opacity Classification System III; NS, nuclear sclerosis.

## Related literature

Our comprehensive search for similar studies revealed 37 studies (Table 1) [8, 10–45]. Given the long study period (27 years), the cohorts differed in terms of phacoemulsification machines and parameters: for example, while longitudinal mode was used in the older studies and still predominated in the more recent studies, torsional and transversal modes have become increasingly favored (S1 Table). Reporting of eligibility criteria was often minimal: most only reported exclusion of pre-existing corneal pathologies. Few studies specified a high preoperative ECD but mean preoperative ECD was >2000 cells/mm$^2$ in nearly all cases (S2 Table). Cataract density was not specified in a fifth of studies and the remaining cohorts differed in cataract density: ours and eight others mostly had moderate-hard cataracts, two cohorts had hard cataracts only, and the remainder had primarily mild and/or moderate cataracts.

Of the 37 studies, 15 were RCTs and the rest were observational studies. Eye number ranged from 23 to 859 but 73% of studies had <100 eyes. Only four studies had samples sizes exceeding ours (n = 275). ECL was measured between 1 and 12 months in all studies except one, which assessed ECL >10 years after surgery. Most studies examined only one or two of the

**Table 3. Univariate analysis of the relationship between endothelial cell loss at month 3 and demographic, clinical, and operative variables.**

| Variable | n | Average±SD ECL | Correlation coefficient | p* |
|---|---|---|---|---|
| Age | 275 | | 0.27 | **<0.001** |
| Sex | | | | 0.35 |
| Female | 159 | 14%±15% | | |
| Male | 116 | 12%±15% | | |
| Cataract density | | | | **0.04** |
| NS1/2 | 48 | 8%±12% | | |
| NS3 | 134 | 15%±15% | | |
| NS4/5 | 91 | 13%±15% | | |
| ACD | 275 | | -0.01 | 0.81 |
| Preoperative CCT | 275 | | 0.04 | 0.49 |
| 2-Hour postoperative CCT | 234 | | 0.06 | 0.38 |
| Preoperative ECD | 275 | | -0.05 | 0.42 |
| Surgical technique | | | | 0.76 |
| Subluxation | 144 | 13%±14% | | |
| Divide-and-conquer | 131 | 13%±16% | | |
| Surgery time | 275 | | 0.08 | 0.17 |
| Ultrasound power | 275 | | 0.25 | **<0.001** |
| APT | 275 | | 0.24 | **<0.001** |
| EPT | 275 | | 0.36 | **<0.001** |

*p values were determined by null hypothesis of zero correlation for Pearson coefficients, Student *t*-tests for two mean comparisons, or ANOVA for three mean comparisons.

ACD, anterior chamber depth; APT, absolute phaco time; CCT, central corneal thickness; ECD, endothelial cell density; EPT, effective phaco time; NS, nuclear sclerosis.

nine patient/surgery variables in the present study. While most involved univariate analyses alone, 13 conducted multivariate analyses (Table 1).

## Overall 3–12-month ECL

The mean 3-month ECL of our cohort was 13%. To date, post-phacoemulsification ECL has been reviewed in specific settings only: meta-analyses have compared diabetics to non-diabetics [68], biaxial microincision phacoemulsification to conventional phacoemulsification [69], FLACS to conventional phacoemulsification [70–73], longitudinal to torsional phacoemulsification [74], and various OVDs [75]. These analyses either included limited numbers of studies (n = 3–9) [68–71, 73, 74] or expressed ECL as absolute cell counts [71, 72, 75]. The ECL% of large numbers of relatively unselected cohorts has not been reviewed. Therefore, we determined the average ECL of the Table 1 cohorts with 3–12-month ECL data (S2 Table). The 3–12-month timepoint was chosen because longitudinal studies suggest that ECL has stabilized by 3 months [12, 16, 18, 33, 52]. Two studies that only examined very hard cataracts [20, 28] were excluded. The mean±SD ECL of the 21 studies was 12±5% (range, 4–21%). RCTs tended to have lower ECL (10±4%; range, 5–17%) than other study types (13±6%; range, 4–21%) (Fig 1). Thus, the ECL of our cohort (13%) is consistent with that in similar cohorts.

However, the wide and enduring differences between studies in terms of ECL over the last 27 years are remarkable (Fig 1). This likely reflects the cumulative contribution of many ECL-shaping factors, including study factors (*e.g.* additional eligibility criteria such as excluding eyes with operative/postoperative complications), patient factors (*e.g.* cataract grade, age, and

**Table 4. Multiple linear regression analysis of the ability of pre/perioperative variables to predict endothelial cell loss at 3 months.**

| Variable | DDL | Parameter estimates | Standard deviation | t value | Pr>F |
|---|---|---|---|---|---|
| Age | 1 | 0.002 | 0.001 | -1.98 | **0.0493** |
| Sex | 1 | 0.023 | 0.019 | -1.22 | 0.223 |
| Cataract hardness | 1 | 0.017 | 0.014 | -0.13 | 0.899 |
| ACD | 1 | 0.026 | 0.024 | -1.10 | 0.274 |
| Preoperative ECD | 1 | 0.00004 | 0.00003 | -1.13 | 0.261 |
| Surgical technique | 1 | -0.007 | 0.022 | 0.31 | 0.754 |
| EPT | 1 | 0.012 | 0.003 | -3.69 | **0.0003** |
| Surgery time | 1 | 0.004 | 0.01 | -0.40 | 0.689 |
| 2-hour postoperative CCT | 1 | 0.00005 | 0.0001 | -0.70 | 0.487 |

ACD, anterior chamber depth; APT, absolute phaco time; CCT, central corneal thickness; ECD, endothelial cell density; EPT, effective phaco time.

comorbidities), and surgical factors (*e.g.* phacoemulsification machine, technique, incision length, OVD, irrigant, and/or intraocular lens).

## Relationship between ECL and pre/perioperative variables

Each of the nine independent variables are discussed below.

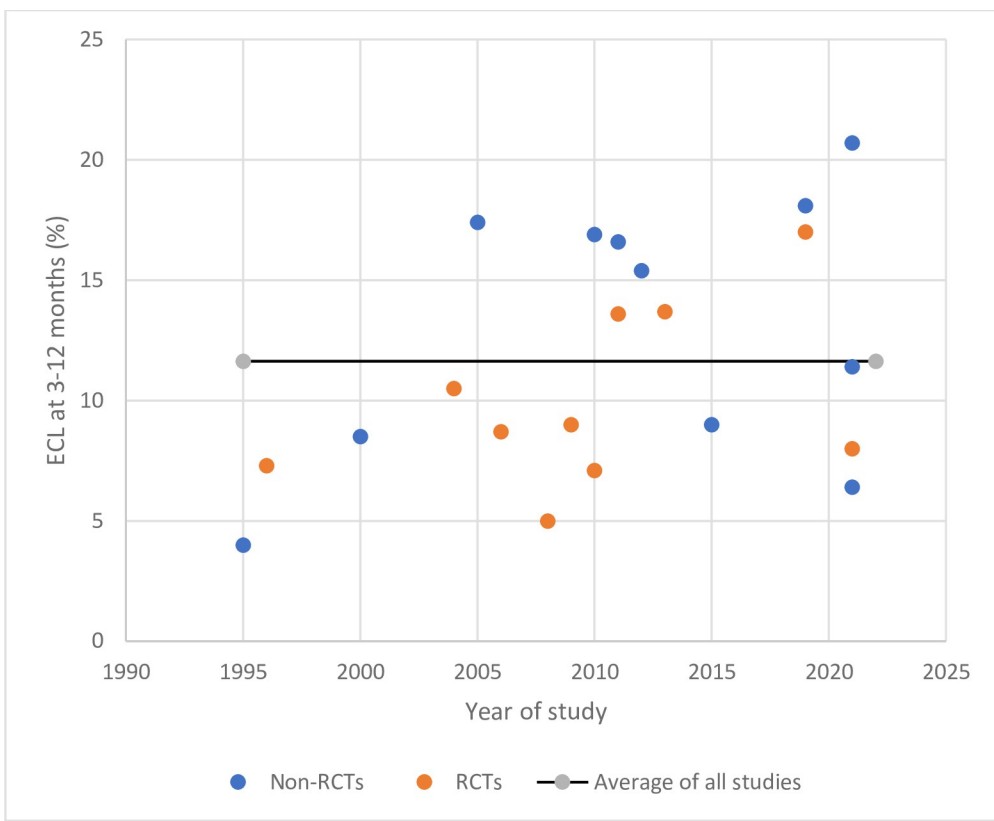

**Fig 1. ECL 3–12 months after phacoemulsification in the literature in 1995–2022.** The studies that were included had cohorts that were similar to ours.

**EPT.** In the literature, the ultrasonic energy expended during phacoemulsification is expressed as two different but related values, namely, EPT and cumulative dissipated energy (CDE). These metrics are sometimes confused [14, 24, 76, 77]. EPT is automatically calculated by Stellaris (our phacoemulsification machine; Bausch & Lomb) and Sovereign WhiteStar (Advanced Medical Optics). It is defined as APT in seconds × average ultrasound power % [56] and is expressed in seconds. The newer WhiteStar Signature system of Advanced Medical Optics, which employs the torsional Ellips FX handpiece, provides EFX: EFX is EPT × a transversal movement coefficient and is also expressed in seconds [23]. CDE is automatically calculated by the Infiniti and Centurion Vision Systems of Alcon Laboratories. It is defined as phaco time in minutes × average phaco power % divided by 100. If the torsional mode is used, the calculation result is multiplied by 0.4 to reflect the reduced heat [23, 27, 70]. CDE is sometimes expressed as %-seconds [70, 78] but is mostly cited without units.

Our study showed that EPT independently predicted ECL. The association between EPT/CDE (or their components) and ECL was observed early in the literature: Dick et al. reported in 1996 that ECL correlated with phacoemulsification energy [44]. Table 1 shows that 12 of 15 univariate analyses have found correlations between EPT/CDE and ECL. Moreover, EPT/CDE independently predicted ECL in four of seven multivariate analyses. EPT/CDE may directly induce ECL by increasing temperatures locally and generating destructive free radicals. It may also promote ECL indirectly: as a measure of phacoemulsification time, it correlates with the intensity of the damaging mechanical forces during phacoemulsification, including trauma from surgical instruments, cataract-fragment ricocheting, and irrigation/aspiration and therefore fluid turbulence [43, 79–83].

**Cataract density and other factors shape the relationship between EPT/CDE and ECL.** Cataract density associated with ECL on univariate analysis in our study. This was also observed in 12 of 15 univariate analyses in the literature (Table 1). This cataract density-ECL relationship is widely thought to be secondary to the effect of EPT/CDE on ECD: more mechanical energy is needed to fragment harder cataracts. Indeed, in our study, cataract density correlated significantly with EPT on univariate analysis (r = 0.25, p<0.0001) and the univariate association between ECL and cataract hardness disappeared on multivariate analysis. Notably, however, the literature shows the opposite pattern: all five multivariate analyses that included both cataract density and EPT/CDE found that cataract density, not EPT/CDE, predicted ECL [36, 39, 40, 43] (Table 1). This discrepancy may reflect collinearity between cataract density and EPT/CDE: many studies show that EPT/CDE correlates with cataract density. This correlation is observed for all cataract grading systems: r ranges from 0.37 to 0.98 with LOCSIII [14, 22, 27, 54, 84–88], 0.40 to 0.92 with Pentacam Scheimpflug imaging [84–87, 89–91], and 0.43 and 0.8 with newly proposed systems [88, 92] (S4 Table and S1 Fig). A rule of thumb is that r >0.7 can signal collinearity between predictor variables, which can create type I and II errors [93]. Since r for cataract density–EPT/CDE correlations can exceed 0.7, the discrepant five multivariate analyses may have suffered from collinearity. Indeed, one reported an r value of 0.69 [36]. By contrast, collinearity was absent in our study: r was 0.25 and none of our predictor variables showed collinearity on Variance Inflation Factor and condition index tests. Thus, the effect of cataract density on ECL may indeed be secondary to that of EPT/CDE.

It is interesting that the cataract density–EPT/CDE relationship varies markedly between cohorts (r = 0.24–0.98), including when the cataract-density range is broad (NC/NO 0–7) [54, 85–88] (S4 Table and S1 Fig). This suggests that while cataract density can greatly shape the ECL-inducing effect of EPT, this may not always be the case. This in turn suggests that (i) EPT/CDE has an inherent capacity to destroy the corneal endothelium that can be distinguished from the effect of harder cataracts, and (ii) surgical/patient conditions can ameliorate/promote the effect of cataract hardness on EPT/CDE. Since targeting these conditions could

reduce cataract density-related ECL, it is of interest to identify them. An example of surgical conditions is incision length: the multivariate analysis of a RCT by Lee et al. showed that while ECL was unchanged over LOCSIII grade 2–4 cataracts in the long-incision (2.2 mm) group, it rose markedly with grade in the short-incision (1.8 mm) group. Concomitantly, CDE increased with increasing cataract grade in both groups but rose much more per grade in the short incision group [36]. Thus, surgical approaches can shape how strongly cataract grade increases EPT/CDE.

Regarding patient variables, one possibility is ACD: Hwang et al. reported that shallow ACD associates with more ECL in grade 3–4 cataracts than deeper ACDs, but this difference is not observed with grade 2 cataracts [94]. Similarly, when Cho et al. subjected eye subgroups with shallow, moderate, or deep ACD to separate multivariate analyses, 3-month ECL was predicted by cataract grade (but not EPT) in the shallow and deep ACD subgroups and by EPT (but not cataract grade) in the moderate ACD subgroup [33]. Thus, the cataract density/EPT/ECL relationship may be amplified in certain anatomical settings. Other studies also report that ACD or related anatomical variables such as axial length associate with ECL [11, 20, 33, 34, 42, 58, 62–64, 94]. However, this is not always observed [12, 15, 30, 37, 38, 40, 95]. Table 1 also shows that the ACD–ECL association was only observed in three of 10 univariate and one of four multivariate analyses. Interestingly, while ACD also failed to predict ECL in our study, we found recently that shallower ACD tended to predict greater ECL (Beta = 0.09; p = 0.11) in the DAC arm of our PERCEPOLIS cohort; interaction analyses then indicated that shallower ACDs associated with more ECL in soft cataracts in the DAC arm (Beta = -0.29; p = 0.18) [96]. By contrast, these relationships and interactions were not observed for the subluxation arm of our cohort [96], nor did the whole cohort demonstrate an ACD:cataract-grade interaction in terms of ECL (Beta = -0.006; p = 0.85). Thus, surgical variables such as incision length or cataract-removal technique may work together with patient variables like ACD to shape the cataract density/EPT/ECL relationship. This may help explain why the associations between ECL and surgical/patient variables vary so much in the literature.

**Age and preoperative ECD.** Since older people already have fewer corneal endothelial cells [97], age is a risk factor for PBK. We observed that older age was the other significant predictor of ECL on multivariate analysis. The Table 1 studies also reported this association in five of 12 univariate and four of seven multivariate analyses. Since (i) the preoperative ECD of our patients was ample (2386±286 cells/mm$^2$), (ii) age predicted ECL independently of the other variables (including preoperative ECD), and (iii) we excluded patients with non-cataract ocular diseases, it may be that an older age imposes a fundamental fragility that enhances the deleterious effects of phacoemulsification. This is consistent with the fact that aging associates with 0.3–0.6% ECL per year [98, 99], possibly due to senescence and cumulative environmental stress-induced oxidative damage [100, 101].

The possibility that endothelial fragility can promote postoperative ECL is supported by phacoemulsification studies on preoperative ECD. While ECD was not a significant factor in our study, one of four univariate and two of five multivariate analyses in the Table 1 studies reported that lower preoperative ECDs predicted higher ECL (Table 1), even though all cohorts had normal preoperative ECDs (S2 Table). Similarly, Chen et al. showed that ECL was 15% in eyes with >2000 cells/mm$^2$ ECD but 29% in eyes with 1000–2000 cells/mm$^2$ ECD [60]. Hayashi et al. also found that eyes with 500–1000 cells/mm$^2$ ECD tended to show more 3-month ECL than eyes with >2300 cells/mm$^2$ ECD (5.1% *vs.* 4.2%, p = 0.15) [59].

The endothelial fragility suggested by the associations of ECL with age and preoperative ECD could at least partly reflect additional ECL risk factors such as diabetes mellitus: diabetes promotes phacoemulsification-induced ECL in eyes with both low [58] and normal preoperative ECD [68]. Notably, Chen et al. [60], Hayashi et al. [59], and the Table 1 multivariate

studies that found preoperative ECD predicted ECL [16, 35] all included diabetics in their cohorts, whereas our study and three of the six studies that failed to find this association excluded diabetics [12, 15, 36] (the other three studies did not detail their eligibility criteria [39–41]).

**Early postoperative CCT.** Phacoemulsification induces corneal swelling: CCT rises by 5–18% in the first day [14, 18, 26, 28, 32, 39]. In eyes without ocular diseases such as Fuchs disease, this swelling drops sharply over the next few days and CCT returns to preoperative levels by 1–3 months [12, 14, 17, 18, 26, 28, 31, 32, 39]. In 1988, a prospective study noted that CCT change 2 and 5 days after cataract surgery correlated positively with ECL at 1 and 6 months [102]. Four cohort studies (including one from our group [18]) then found that 2-hour/1-day CCT correlated with ECL at 1 and 3 months and even 10 years later (r = 0.4–0.56) [15, 18, 29, 39]. However, on multivariate analysis, day-1 CCT predicted ECL in one study (at >10 years) [15] but not another (at 3 months) [39] (Table 1). Our present study also failed to find any association between 2-hour CCT and 3-month ECL. Further studies are needed to determine whether early CCT could be a useful marker of later ECL.

**Other variables.** Sex, surgical technique, and surgery time did not associate with ECL in our study or the literature (Table 1).

## Improving phacoemulsification research

Our study showed that, consistent with the literature, older age and higher EPT predicted 3-month ECL Since EPT/CDE is a modifiable factor, unlike age, a key research focus over the past decades has been to reduce EPT/CDE. Many approaches have been implemented or proposed, including the use of chopping techniques rather than DAC [103], OVDs [75, 104], pulse, burst, or microburst rather than continuous approaches [105–107], torsional or transversal rather than longitudinal phacoemulsification [74, 106, 108, 109], active or hyper-pressurized fluidics rather than passive fluidics [110], and FLACS [70–73]. These changes together with advances in phacoemulsification machine technology have significantly improved ECL: the first case series on ECL after phacoemulsification, which was reported in 1978 by Sugar et al., had an ECL of 34% [111] whereas the ECL over the last 20 years averages 12% (Fig 1).

However, the research on ECL has several important limitations that should be addressed to allow further improvements in phacoemulsification outcomes. One is that many studies use EPT/CDE as a surrogate of ECL. This is inappropriate for two reasons: (i) there are many RCTs where an intervention significantly affects EPT/CDE but not ECL [8, 112, 113] and vice versa [30, 114, 115]. (ii) EPT accounted for only 5.1% of the total ECL variation in our cohort. Baradaran et al. also observed that it was a weak predictor [35]. Notably, however, another study found CDE had a large effect [16]; this disparity is likely due to the covariates that were included and different cohort characteristics and consolidates the crucial importance of including other patient/surgical factors. Another significant limitation of ECL research is that most studies do not assess the impact of an intervention/factor on EPT/CDE and ECL in different cataract densities: they either determine mean EPT/CDE and ECL across a range of cataract densities or focus on mild/moderate or hard cataracts only. Some do not even report cohort cataract density. Our analysis shows that cataract density can greatly shape the ECL-inducing effects of EPT/CDE. Thus, not considering cataract hardness could lead, for example, to an intervention being used with soft cataracts when in fact it induces more ECL with such cataracts. Considering cataract density will also help isolate the effect of cataract hardness from the inherent cataract-independent capacity of EPT/CDE to destroy the corneal endothelium. This could be useful for identifying targetable mechanisms by which EPT/CDE destroy corneal endothelial cells.

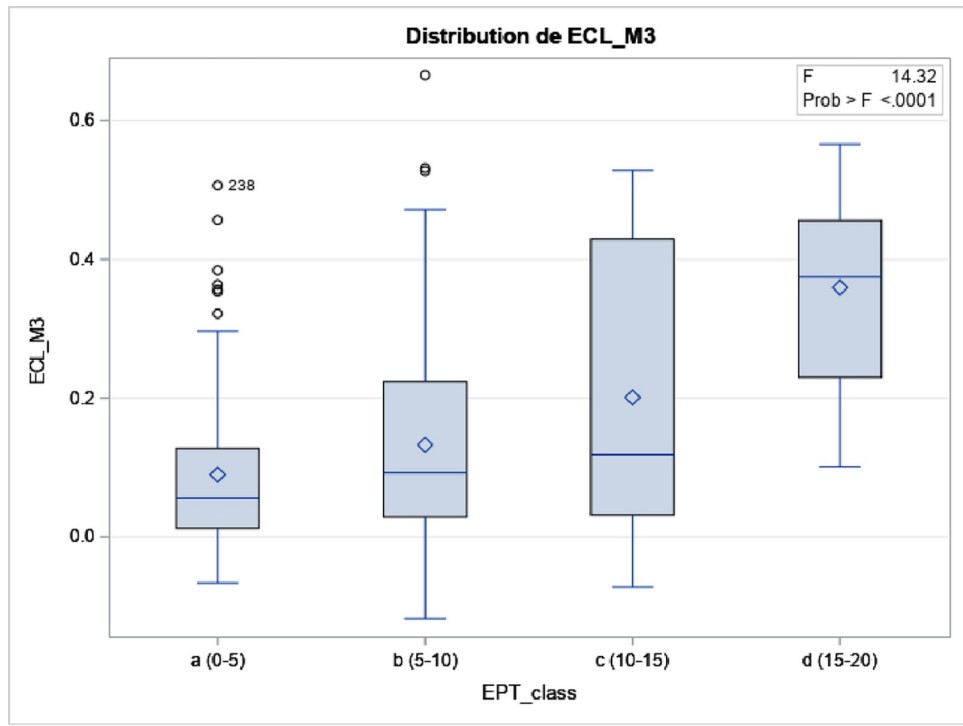

**Fig 2. Effect of increasing EPT on ECL.** EPT was categorized as 0–4.99, 5–9.99, 10–14.99, and 15–19.99 seconds.

An important consideration in such research is the cataract grading system. There are now more than 56 proposed grading systems [116], all of which have advantages and disadvantages [92]. However, a recent comprehensive comparison of several methods showed that LOCSIII correlated best with CDE [85]. Moreover, while LOCSIII does depend on reference photographs, it is comprehensive, detailed, and widely used in the literature [85].

Another limitation in phacoemulsification research is insufficient definition of the patient/eye/surgical characteristics of the cohort. This is common (S1 and S2 Tables). Our present study suggests that interactions between patient/surgery factors (e.g. cataract grade, eye anatomy, and cataract-removal technique/incision length) shape EPT/CDE and the resulting ECL. To identify the precise constellations of patient/surgical factors that promote ECL, further research with well-characterized cohorts is needed.

**Non-linear relationship between EPT/CDE and ECL.** We noted that EPT and ECL tended to have a non-linear relationship: ECL was low until it exceeded 15 seconds, after which ECL rose sharply ($p<0.0001$) (Fig 2). Similarly, an RCT comparing FLACS and conventional phacoemulsification showed that ECL doubled when CDE exceeded 10 [16]. This may partly reflect a non-linear relationship between EPT/CDE and cataract hardness: Davison et al. observed a linear relationship until LOCSIII ~3.7, after which the relationship became exponential [117]. In any case, these findings suggest that surgeons should seek to avoid >15-second EPTs or >10 CDEs.

## Study limitations

This study had several strengths: it was based on RCT data, had a large sample size, conducted a multivariate analysis with nine patient/surgical variables, and the data were presented in the context of the large and complex research field. However, 82% of the cataracts in our study

had NS3–4 density. Thus, the low frequencies of NS1, NS2, and NS5 cataracts may have precluded us from detecting variables that contribute significantly to ECL in such cases. Moreover, all surgeries were conducted by a single experienced surgeon: thus, our data may not be generalizable to other settings. Finally, we did not examine the influence of other potential ECL-shaping variables, including diabetes, other ocular anatomical variables, the amount of fluid used, the effect of intra-camerular products such as trypan blue, and surgeon experience. To address this, we are currently conducting a large-scale prospective study assessing the impact of these and other variables on ECL, particularly in the context of different cataract grades.

## Supporting information

**S1 Fig. Plot of correlation coefficients in the literature.** CD, cataract grade; EPT, effective phaco time.
(DOCX)

**S1 Table. Objective, number of eyes, technical details, eligibility criteria, and ECL reported by the studies shown in Table 1.**
(DOCX)

**S2 Table. Summary of exclusion criteria used for the studies shown in Table 1 (where applicable, control group only).**
(DOCX)

**S3 Table. Comparison of eyes for which 2-hour CCT data were (n = 234) and were not (n = 41) available.**
(DOCX)

**S4 Table. Correlation coefficients for the relationship between cataract density and EPT in the literature.**
(DOCX)

## Author Contributions

**Conceptualization:** Jean-Marc Perone, Christophe Goetz.

**Data curation:** Jean-Marc Perone, Marie-Soline Luc, Jean-Charles Vermion, Grace Gan, Christophe Goetz.

**Formal analysis:** Yinka Zevering, Christophe Goetz.

**Funding acquisition:** Jean-Marc Perone.

**Investigation:** Jean-Marc Perone, Marie-Soline Luc, Yinka Zevering, Jean-Charles Vermion, Grace Gan.

**Methodology:** Jean-Marc Perone, Christophe Goetz.

**Project administration:** Jean-Marc Perone.

**Resources:** Jean-Marc Perone, Christophe Goetz.

**Software:** Christophe Goetz.

**Supervision:** Jean-Marc Perone.

**Visualization:** Yinka Zevering.

**Writing – original draft:** Marie-Soline Luc, Yinka Zevering.

**Writing – review & editing:** Jean-Marc Perone, Marie-Soline Luc, Yinka Zevering, Jean-Charles Vermion, Grace Gan, Christophe Goetz.

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
