## [Decision Letter · Decision Letter 0]

8 Oct 2023

PONE-D-23-24547Post-hoc trial analysis and narrative review of factors that predict corneal endothelial cell loss after phacoemulsification: tips for improving cataract surgeryPLOS ONE

Dear Dr. Perone,

Thank you for submitting your manuscript to PLOS ONE. After careful consideration, we feel that it has merit but does not fully meet PLOS ONE’s publication criteria as it currently stands. Therefore, we invite you to submit a revised version of the manuscript that addresses the points raised during the review process.

We look forward to receiving your revised manuscript.

Kind regards,

Georgios Labiris, MD, PhD

Academic Editor

PLOS ONE

Journal Requirements:

2. Please ensure that you have specified a) Did participants provide their written or verbal informed consent to participate in this study?

Reviewers' comments:

Reviewer's Responses to Questions

**Comments to the Author**

1. Is the manuscript technically sound, and do the data support the conclusions?

Reviewer #1: Yes

Reviewer #2: Yes

Reviewer #3: Yes

2. Has the statistical analysis been performed appropriately and rigorously? 

Reviewer #1: Yes

Reviewer #2: Yes

Reviewer #3: I Don't Know

3. Have the authors made all data underlying the findings in their manuscript fully available?

Reviewer #1: Yes

Reviewer #2: Yes

Reviewer #3: Yes

4. Is the manuscript presented in an intelligible fashion and written in standard English?

Reviewer #1: Yes

Reviewer #2: Yes

Reviewer #3: Yes

5. Review Comments to the Author

Reviewer #1: This manuscript seeks to identify predictive factors for endothelial cell loss after cataract surgery, a very important and clinically relevant topic. However, this topic has previously been extensively studied. This study does indeed have some strengths over other previously published manuscripts, with improvements to the methodology and study design. However this study does not change the overall conclusion and does not make any meaningful contribution to the literature, as older age and dense cataract are already well-known and widely accepted predictors of endothelial cell loss.

Line 100: It does not make sense to say exclusion criteria included "preexisting cornea" and "intraocular pressure," as these are just parameters. Do the authors mean pre-existing corneal pathology and elevated intraocular pressure?

Line 101: Do the authors mean endothelial cell density LESS than 1500 cells/mm2? ">" should be changed to "<"

Line 113: How did the surgeon decide to use DAC vs subluxation?

Line 143: What were the technical reasons why surgical technique was altered?

Lines 394-405: This discussion about hypothetical conditions needs to be removed or significantly altered. This discussion is meaningless because conditions A-E have not been defined. Moreover, hypothetical results have no place in a manuscript like this.

Reviewer #2: The investigators take care to note that this is a retrospective or post hoc look of a larger well designed cilnical trial with what appears to be an adequate sample size. The multivariate approach appears to be somewhat convincing in that only an older age (beta=0.2%, p=0.049) and higher EPT (beta=1.2%, p=0.0002) predicted 3-month ECL. Cataract density was significant on univariate (p=0.04) but not multivariate analysis. Study limitations are noted including the fact that they are currently conducting a large-scale prospective study assessing the impact of these and other variables on ECL, particularly in the context of different cataract grades.

All the statistical tools were in place for addressing this post hoc concern and clearly the investigators demonstrated that sufficient leads are in place to confirm these results. Apparently available data was collected as seen in Tables 1 and 2 to investigate the preoperative, operative and demographic data. Univariate and multivariate results are demonstrated clearly in tables 3 and 4.

It appears that the three month time is chosen for investigating the ECL. One wonders if that is the only time point considered in this analysis or other time points could have been considered to investigate a time dependent influence of the variables on the ECL. This is a non clinical statistical reviewer wondering such.

Reviewer #3: Thank you for the opportunity to review this paper by Perone and colleagues, In general, it is a very well written paper about an important but also complex topic, that of the effects of phacoemulsification on endothelial cell loss.

I believe the methods, analysis and the results of the paper are valid, and in line with the current literature that the authors have extensively reviewed.

My main concern is directed towards the lengthy discussion in the paper, as well as some of the suggestions made on future research. As a clinical research paper, I would say that the discussion is far too long and veers into the territory of being a review paper on its own. The authors may wish to consider discussing their own personal results, and conjecture on why they do or do not affect 3 month ECL, without referencing all of past research that has been done. The actual review of the literature is very comprehensive, but may be better served as a separate paper in another journal.

One of the suggestions made by the authors in improving the research in this field is to clearly define and document the cataract density of the study cohort. They state that this is important as “Our analysis shows that cataract density can greatly shape the ECL-inducing effects of EPT/CDE”. While I believe that is indeed important, the results of this study do not directly demonstrate that association between cataract density and ECL on multivariate analysis. Perhaps they drew this conclusion based on the 5 previous metanalyses that did show a correlation (lines 284-285)? Even so, even if there were an association between cataract density and EPT, the overall effect may yet be small, as their study results only found that ECL increased 1.2% for every second longer EPT (line 210).

In terms of the limitation of the study, it would be prudent to address the fact that this study was a posthocc analysis of a trial that was primarily looking at the differences between two surgical techniques. As such, the sample size of this study, while large, may not have been sufficiently powered to identify variables that contribute to ECL loss. An example would again be cataract density, where there were very few cases with very soft (NS1+1) and very dense (NS5+) due to the convenience sampling. Finally, this is essentially the results of a single, very experienced surgeon, which may again result in systematic biases in the results.

6. PLOS authors have the option to publish the peer review history of their article (what does this mean?). If published, this will include your full peer review and any attached files.

Reviewer #1: No

Reviewer #2: No

Reviewer #3: No

---

## [Author Response · Author response to Decision Letter 0]

18 Nov 2023

Point-by-point Response to Reviewer Comments

Editor Comments

Reply: We have uploaded a rebuttal letter, a revised manuscript with Track Changes, and a revised manuscript without tracked changes.

 and 

Reply: We have changed the formatting of the paper and file names according to the instructions in the links.

2. Please ensure that you have specified a) Did participants provide their written or verbal informed consent to participate in this study?

Reply: All subjects provided written informed consent before randomization. This was specified in the Study design and ethics section. (Line 112)

Reply: The datasets generated during and/or analyzed during the current study are not publicly available according to French Law No. 2018-493 of June 20, 2018 on the protection of personal data (The General Data Protection Regulation (Regulation (EU) 2016/679) (GDPR: article 9) but are available from the Clinical Research Support Platform (Plateforme d’Appui à la Recherche Clinique [PARC]) of the Regional Central Hospital (CHR) of Metz-Thionville on reasonable request (email: projetrecherche@chrmetz-thionville.fr, tel: +33 3 87 17 98 82). All non-archived data is subject to daily backups while all archived data is subject to duplicate storage at two different sites. This data processing is compliant with a baseline reference methodology (MR-004) to which the CHR MetzThionville signed a compliance commitment on October 8, 2018.

Reply: We removed the affiliation “3 LORIA, STID Department, IUT of Metz, Lorraine University, Metz, Grand Est, France” and indicated the affiliation for Dr. Zevering

Reply: We have provided the captions for Supplementary Tables S1–S4 at the end of the manuscript.

Reviewer Comments

Reviewer #1: 

This manuscript seeks to identify predictive factors for endothelial cell loss after cataract surgery, a very important and clinically relevant topic. However, this topic has previously been extensively studied. This study does indeed have some strengths over other previously published manuscripts, with improvements to the methodology and study design. However this study does not change the overall conclusion and does not make any meaningful contribution to the literature, as older age and dense cataract are already well-known and widely accepted predictors of endothelial cell loss.

Reply: Thank you very much for taking the time to review and comment on our manuscript. We also appreciate your positive comments about our methodology and study design. We believe that our post-hoc trial data in the context of our narrative review add substantially to the field because:

• There is significant confusion in the field about the relationship between effective phaco time (EPT) and cataract density and how this affects endothelial-cell loss (ECL): multivariate analyses that included both EPT and cataract density showed that cataract density, not EPT, predicted ECL even though it is widely thought that the cataract density-ECL relationship is secondary to the effect of EPT on ECL. Our narrative review showed that this confusion likely reflects collinearity between cataract density and EPT and the inappropriate inclusion of these two variables in the same multivariate analysis.

• In unraveling this confusion, we noted that the correlation between cataract density and EPT varies markedly between cohorts: even in cohorts with a broad range of cataract densities, the cataract density:EPT correlation ranges from low (e.g. r=0.25 in our study) to very high in others (e.g. r=0.98). We were also struck by the finding of Lee et al. that a cataract density-EPT-ECL relationship only emerged when short-incisions were used rather than long-incisions. We were also interested in the study of Cho et al. that showed the cataract density-EPT-ECL relationship varied with ACD. These findings suggested to us that there may be patient and surgical conditions that could alter the cataract density-EPT relationship. 

• These findings in turn suggest that it will be necessary to carefully dissect the cataract density-EPT-ECL relationship to find patient and surgical factors that reduce ECL in phacoemulsification. To our knowledge, these complex links between patient/surgical factors and ECL have never been comprehensively synthesized in the literature.

• There are also many studies that do not find that older age predict ECL: only four of seven previous multivariate analyses found that age predicted ECL. Our narrative review suggests that this may reflect, at least in part, endothelial fragility that is due to additional ECL risk factors such as diabetes. This highlights the importance of including endothelial fragility-related variables in multivariate analyses searching for patient/surgical factors that associate with increased ECL.

Thus, our study elucidates a very large, highly inconsistent, and complex field and points the way to improving research on factors that predict phacoemulsification-induced ECL. Since this analysis has significantly informed our own ongoing research in this field, we believe it will be useful to others as well.

Line 100: It does not make sense to say exclusion criteria included "preexisting cornea" and "intraocular pressure," as these are just parameters. Do the authors mean pre-existing corneal pathology and elevated intraocular pressure?

Reply: Yes, we used semi-colons to signify that exclusion criteria included preexisting cornea pathology, IOP pathology, and posterior-segment pathology. We understand that it is confusing to read and made it clearer as follows:

“Exclusion criteria were: white/brown cataracts; insulin-dependent diabetes or diabetic retinopathy; preexisting cornea pathology; intraocular-pressure (IOP) pathology; posterior-segment pathology; preoperative endothelial-cell density (ECD) >1500 cells/mm2; pregnancy;...” (Lines 120–122)

Line 101: Do the authors mean endothelial cell density LESS than 1500 cells/mm2? ">" should be changed to "<"

Reply: Thank you for picking this up. Yes, it should be <1500 cells/mm2. We made the change. (Line 122)

Line 113: How did the surgeon decide to use DAC vs subluxation?

Reply: The patients were randomized to either of the two treatment arms. We made this clearer as follows:

“Briefly, the cohort consisted of a convenience series of 292 adult patients who had a nuclear (NO1–NO4; NC1–NC4), cortical (C1–C5), or posterior subcapsular (P1–P5) cataract, as determined by using the Lens Opacities Classification System (LOCS)III classification [51], and a best spectacle-corrected visual acuity (BSCVA) of >+0.2 logMAR and were randomized to undergo either subluxation (n=148) or DAC (n=144).” (Lines 115–120)

Line 143: What were the technical reasons why surgical technique was altered?

Reply: To address this point, we added the following text to the Methods:

“The per-protocol trial data were analyzed: 15 of the 292 phacoemulsifications had been converted to the other method for technical reasons (an overly soft core, poor pharmacologically induced pupillary dilation, significant anterior chamber narrowness, and severely hard crystalline lens).” (Lines 165–168) 

Lines 394-405: This discussion about hypothetical conditions needs to be removed or significantly altered. This discussion is meaningless because conditions A-E have not been defined. Moreover, hypothetical results have no place in a manuscript like this.

Reply: We agree and have deleted all of the texts and Fig. 2 about the hypothetical cataract grade-EPT relationships.

Reviewer #2: 

The investigators take care to note that this is a retrospective or post hoc look of a larger well designed cilnical trial with what appears to be an adequate sample size. The multivariate approach appears to be somewhat convincing in that only an older age (beta=0.2%, p=0.049) and higher EPT (beta=1.2%, p=0.0002) predicted 3-month ECL. Cataract density was significant on univariate (p=0.04) but not multivariate analysis. Study limitations are noted including the fact that they are currently conducting a large-scale prospective study assessing the impact of these and other variables on ECL, particularly in the context of different cataract grades.

All the statistical tools were in place for addressing this post hoc concern and clearly the investigators demonstrated that sufficient leads are in place to confirm these results. Apparently available data was collected as seen in Tables 1 and 2 to investigate the preoperative, operative and demographic data. Univariate and multivariate results are demonstrated clearly in tables 3 and 4.

Reply: Thank you very much for taking the time to read and comment on our manuscript, and for your positive comments.

It appears that the three month time is chosen for investigating the ECL. One wonders if that is the only time point considered in this analysis or other time points could have been considered to investigate a time dependent influence of the variables on the ECL. This is a non clinical statistical reviewer wondering such.

Reply: Yes, it is a good point that the relationships between the variables could potentially change over time. However, we had to choose the 3-month point because our 12-month data were complicated by a large loss-to follow-up (29%). By contrast, only 6% were lost to follow-up at 3 months. To address this point, we added the following text to the Methods:

“ECD measured 3 months after surgery was used to calculate ECL at 3 months relative to baseline. The 3-month timepoint was chosen for analysis rather than the 12-month timepoint in our trial because loss to follow-up was 6% at 3 months and 29% at 12 months.” (Lines 151–154)

Reviewer #3: 

Thank you for the opportunity to review this paper by Perone and colleagues, In general, it is a very well written paper about an important but also complex topic, that of the effects of phacoemulsification on endothelial cell loss.

I believe the methods, analysis and the results of the paper are valid, and in line with the current literature that the authors have extensively reviewed.

Reply: Thank you very much for taking the time to review and comment on our paper, and for your positive comments. 

My main concern is directed towards the lengthy discussion in the paper, as well as some of the suggestions made on future research. As a clinical research paper, I would say that the discussion is far too long and veers into the territory of being a review paper on its own. The authors may wish to consider discussing their own personal results, and conjecture on why they do or do not affect 3 month ECL, without referencing all of past research that has been done. The actual review of the literature is very comprehensive, but may be better served as a separate paper in another journal.

Reply: Indeed, our manuscript describes a posthoc trial analysis that is interpreted in the context of a very large, complex, and confusing field in the form of a narrative review. We decided to take this approach because although our multivariate analyses did not yield particularly novel predictive factors for endothelial-cell loss (ECL), our comprehensive review and analysis of the literature revealed important aspects about the cataract density-EPT-ECL relationship that should be taken into account when searching for patient/surgical factors that influence ECL after phacoemulsification. We recognize that our approach – a narrative review centered on our posthoc analysis – is somewhat unorthodox but feel that it will help illustrate how to improve research in this field. Indeed, this analysis has significantly informed our own ongoing research in this field. Therefore, we believe it will be useful to others as well.

 To address this point, we changed the title as follows:

“Narrative review after posthoc trial analysis of factors that predict corneal endothelial cell loss after phacoemulsification: tips for improving cataract surgery research” (Lines 3–6)

One of the suggestions made by the authors in improving the research in this field is to clearly define and document the cataract density of the study cohort. They state that this is important as “Our analysis shows that cataract density can greatly shape the ECL-inducing effects of EPT/CDE”. While I believe that is indeed important, the results of this study do not directly demonstrate that association between cataract density and ECL on multivariate analysis. Perhaps they drew this conclusion based on the 5 previous metanalyses that did show a correlation (lines 284-285)? 

Reply: It is a widely held view in the field that cataract density shapes ECL because more phaco power (EPT) is needed to process a harder lens. This view is borne out by the literature (see new Supplementary Table S4): the average correlation coefficient for the relationship between cataract grade and EPT is r=0.61. This correlation was also significant in our study on univariate analysis (r=0.25; p<0.0001).

What is also notable in the table above is the wide range of correlation coefficients: they run from 0.25 in our study to 0.98 (see new Supplementary Fig S1). These disparities do not reflect different cataract grade distributions since most studies included a wide spread of grades. 

This variation in cataract grade-EPT relationship strength caught our eye when we read the RCT of Lee et al. that compared long and short corneal incisions (ref 29): they observed a cataract density-EPT-ECL relationship with short incisions but not long incisions (see shots of the Lee et al. figures in the attached Response to Reviewers document at the end of this submission). 

Thus, these data together with other studies detailed in the manuscript led to three conclusions: (1) “cataract density can greatly shape the ECL-inducing effects of EPT/CDE” but (2) this effect can be modified by surgical and patient (e.g. ACD) factors; and (3) therefore, to identify modifiable factors that shape ECL, it is essential to consider cataract density, which is often barely (or not) detailed in the literature (e.g. Feng et al. 2022, Lim et al. 2014, and Abell et al. 2013 in new Supplementary Table S4).

To address this important point, we added new Supplementary Table S4 and new Supplementary Fig S1 to the manuscript as well as the underlined Discussion text below:

“This cataract density-ECL relationship is widely thought to be secondary to the effect of EPT/CDE on ECD: more mechanical energy is needed to fragment harder cataracts. Indeed, in our study, cataract density correlated significantly with EPT on univariate analysis (r=0.25, p<0.0001) and the univariate association between ECL and cataract hardness disappeared on multivariate analysis.” (Lines 332–336)

Even so, even if there were an association between cataract density and EPT, the overall effect may yet be small, as their study results only found that ECL increased 1.2% for every second longer EPT (line 210).

We agree that EPT had a relatively small effect in our study. It should be noted that while several multivariate studies in the literature do find that EPT predicts ECL, some do not (see Table 1 of our manuscript). Moreover, one multivariate study (Ref 35) also shows a small effect of EPT whereas another shows a large effect (Ref 16). This variability likely reflects (i) variable cohort characteristics, (ii) the covariates that were included in the multivariate analysis, and (iii) in some cases, the erroneous inclusion of cataract density in multivariate analysis when it is collinear with EPT.

 To address this point, we added the following text to the Discussion:

“However, the research on ECL has several important limitations that should be addressed to allow further improvements in phacoemulsification outcomes. One is that many studies use EPT/CDE as a surrogate of ECL. This is inappropriate for two reasons: (i) there are many RCTs where an intervention significantly affects EPT/CDE but not ECL [8,110,111] and vice versa [30,112,113]. (ii) EPT accounted for only 5.1% of the total ECL variation in our cohort. Baradaran et al. also observed that it was a weak predictor [35]. Notably, however, another study found CDE had a large effect [16]; this disparity is likely due to the covariates that were included and different cohort characteristics and consolidates the crucial importance of including other patient/surgical factors.” (Lines 434–442)

In terms of the limitation of the study, it would be prudent to address the fact that this study was a posthocc analysis of a trial that was primarily looking at the differences between two surgical techniques. As such, the sample size of this study, while large, may not have been sufficiently powered to identify variables that contribute to ECL loss. An example would again be cataract density, where there were very few cases with very soft (NS1+1) and very dense (NS5+) due to the convenience sampling. Finally, this is essentially the results of a single, very experienced surgeon, which may again result in systematic biases in the results.

Reply: These are excellent points. We have added the following text to the Limitations section:

“However, 82% of the cataracts in our study had NS3–4 density. Thus, the low frequencies of NS1, NS2, and NS5 cataracts may have precluded us from detecting variables that contribute significantly to ECL in such cases. Moreover, all surgeries were conducted by a single experienced surgeon: thus, our data may not be generalizable to other settings.” (Lines 478–482)

---

## [Decision Letter · Decision Letter 1]

14 Jan 2024

PONE-D-23-24547R1Narrative review after post-hoc trial analysis of factors that predict corneal endothelial cell loss after phacoemulsification: tips for improving cataract surgery researchPLOS ONE

Dear Dr. Perone,

Thank you for submitting your manuscript to PLOS ONE. After careful consideration, we feel that it has merit but does not fully meet PLOS ONE’s publication criteria as it currently stands. Therefore, we invite you to submit a revised version of the manuscript that addresses the points raised during the review process.

Additional Editor Comments:

-Please replace the subsection title "Statistical analyses" with the title "Statistical analysis"

-Please add a power analysis to determine if the sample size is adequate. 

We look forward to receiving your revised manuscript.

Kind regards,

Georgios Labiris, MD, PhD

Academic Editor

PLOS ONE

Journal Requirements:

Reviewers' comments:

Reviewer's Responses to Questions

**Comments to the Author**

1. If the authors have adequately addressed your comments raised in a previous round of review and you feel that this manuscript is now acceptable for publication, you may indicate that here to bypass the “Comments to the Author” section, enter your conflict of interest statement in the “Confidential to Editor” section, and submit your "Accept" recommendation.

Reviewer #2: All comments have been addressed

Reviewer #3: All comments have been addressed

2. Is the manuscript technically sound, and do the data support the conclusions?

Reviewer #2: Partly

Reviewer #3: (No Response)

3. Has the statistical analysis been performed appropriately and rigorously? 

Reviewer #2: Yes

Reviewer #3: (No Response)

4. Have the authors made all data underlying the findings in their manuscript fully available?

Reviewer #2: Yes

Reviewer #3: (No Response)

5. Is the manuscript presented in an intelligible fashion and written in standard English?

Reviewer #2: Yes

Reviewer #3: (No Response)

6. Review Comments to the Author

Reviewer #2: My concern was with the adequacy of the follow up time. The authors apparently believe that the 3 month time is sufficient. If such is the case then the analysis will follow.

Reviewer #3: (No Response)

7. PLOS authors have the option to publish the peer review history of their article (what does this mean?). If published, this will include your full peer review and any attached files.

Reviewer #2: No

Reviewer #3: No

---

## [Author Response · Author response to Decision Letter 1]

17 Jan 2024

Response to Reviewers

Editor comments

Comment 1:

-Please replace the subsection title "Statistical analyses" with the title "Statistical analysis"

Reply: We have changed the subsection title to “Statistical analysis”. 

Comment 2:

-Please add a power analysis to determine if the sample size is adequate. 

Reply: Before the PERCEPOLIS trial was conducted, a sample size calculation was conducted. This showed that to detect the expected endothelial-cell loss of 5–12% at 1 month (based on the literature at the time), 294 patients had to be recruited. In total, 292 were recruited. This sample size calculation was reported by our published article describing the primary endpoint of the trial (Perone et al. Corneal Endothelial Cell Loss After Endocapsular and Supracapsular Phacoemulsification. 2021. Cornea. 41;714).

However, we believe that the question being asked was, “Was the sample size sufficient for the multivariate analysis in the present study?” To address this, we added the following text and reference 57 to the Statistics section:

“Multiple linear regression analysis was conducted to identify which of nine variables can independently predict ECL at 3 months. Given the cohort sample size (n=275) and the 1 in 10 rule of thumb in multivariate analysis that states one candidate predictor can be studied for every 10 patients,[57] the sample size was ample for this analysis.” Lines 176–180

Comment 3:

Please review your reference list to ensure that it is complete and correct.

Reply: The reference list has been reviewed.

Reviewer Comments

Reviewer 2: My concern was with the adequacy of the follow up time. The authors apparently believe that the 3 month time is sufficient. If such is the case then the analysis will follow.

Reply: We chose to focus on the 3-month timepoint because there was a large loss to follow-up at the later timepoint in the trial (12 months; 93 of 292 lost = 32% loss). Thus, the 12-month cohort may have been selected in some way, meaning they may not have been fully representative of the general phacoemulsification patients. In addition, longitudinal studies show that post-phacoemulsification ECL has stabilized by 3 months: for example, Lass et al. (Am J Ophthalmol. 2019 208;211) showed that ECL was 8.3% and 8.7% at 3 and 24 months, respectively. 

To address this point, we added the following text to the Methods section:

“Although ECD and other data were collected from the PERCEPOLIS cohort at 1, 3, and 12 months, we chose the 3-month data for the present study because there was a large loss to follow-up at 12 months (32%), meaning there may have been some selection of the 12-month patients. Moreover, ECL is generally considered to have stabilized at 3 months [12,16,18,33,52].” (lines 125–128)

---

## [Editor Report · Decision Letter 2]

31 Jan 2024

Narrative review after post-hoc trial analysis of factors that predict corneal endothelial cell loss after phacoemulsification: tips for improving cataract surgery research

PONE-D-23-24547R2

Dear Dr. Perone,

We’re pleased to inform you that your manuscript has been judged scientifically suitable for publication and will be formally accepted for publication once it meets all outstanding technical requirements.

Kind regards,

Georgios Labiris, MD, PhD

Academic Editor

PLOS ONE

---

## [Editor Report · Acceptance letter]

13 Mar 2024

PONE-D-23-24547R2 

PLOS ONE

Dear Dr. Perone, 

I'm pleased to inform you that your manuscript has been deemed suitable for publication in PLOS ONE. Congratulations! Your manuscript is now being handed over to our production team.

Kind regards, 

on behalf of

Dr. Georgios Labiris 

Academic Editor

PLOS ONE